# Investigation of a Gaussian Plume in the Vicinity of an Urban Cyclotron Using Helium as a Tracer Gas

Philippe Laguionie [1,*], Olivier Connan [1], Thinh Lai Tien [2], Sophie Vecchiola [3], Johann Chardeur [1], Olivier Cazimajou [1], Luc Solier [1], Perrine Charvolin-Volta [4], Liying Chen [5], Irène Korsakissok [6], Malo Le Guellec [5], Lionel Soulhac [4], Amita Tripathi [5] and Denis Maro [1]

1 Institut de Radioprotection et de Sûreté Nucléaire (IRSN), PSE-ENV/SRTE/LRC, 50130 Cherbourg-en-Cotentin, France; olivier.connan@irsn.fr (O.C.); johann.chardeur@irsn.fr (J.C.); olivier.cazimajou@irsn.fr (O.C.); luc.solier@irsn.fr (L.S.); denis.maro@irsn.fr (D.M.)

2 Vietnam Agency for Radiation and Nuclear Safety (VARANS), Ha Noi City 122000, Vietnam; ltthinh.varans@gmail.com

3 Institut de Radioprotection et de Sûreté Nucléaire (IRSN), PSE-ENV/SEREN/BERAP, 92260 Fontenay-aux-Roses, France; sophie.vecchiola@irsn.fr

4 Laboratoire de Mécanique des Fluides et d'Acoustique, University of Lyon, CNRS UMR 5509, Ecole Centrale de Lyon, INSA Lyon, Université Claude Bernard, 69134 Ecully, France; perrine.charvolin@ec-lyon.fr (P.C.-V.); lionel.soulhac@ec-lyon.fr (L.S.)

5 FLUIDYN France, 93200 Saint-Denis, France; liying.chen@fluidyn.com (L.C.); malo.leguellec@fluidyn.com (M.L.G.); amita.tripathi@fluidyn.com (A.T.)

6 Institut de Radioprotection et de Sûreté Nucléaire (IRSN), PSE-SANTE/SESUC/BMCA, 92260 Fontenay-aux-Roses, France; irene.korsakissok@irsn.fr

* Correspondence: philippe.laguionie@irsn.fr

**Abstract:** Studies focusing on the radiological impact of fluorine 18 on populations living near to cyclotrons (<200 m) frequently assume normal distribution of atmospheric concentration for simplification purposes. On this basis, Gaussian models are used, despite their limits, as deployment requires little input data and computing resources. To estimate the ability of a Gaussian model to predict atmospheric dispersion in an urban environment, we used helium as a new passive tracer of atmospheric dispersion in the near-field range (<500 m) of the Beuvry hospital cyclotron (France). The atmospheric transfer coefficients *ATC* measured in the field were compared with those modeled using a Gaussian equation. According to the results, helium is an effective tracer of atmospheric dispersion when attempting to determine atmospheric transfer coefficients (*ATC*) downwind of a discharge point. The Briggs-rural, Briggs-urban and Doury Gaussian models underestimate *ATC* and sometimes maximum *ATC* in the prevailing weather conditions during the experiments. By compiling the results of this study with data from the literature, it appears that the maximum *ATC* observed obey a power law as a function of the distance from the discharge point, for distances from the discharge point in excess of 20 m.

**Keywords:** gaussian plume; tracing experiment; helium; cyclotron; urban environment; near-field

## 1. Introduction

Cyclotrons are particle accelerators. They have been used to produce the fluorine 18 required for an ever-increasing number of clinical applications since the start of the 21st century. Fluorine 18 is an emitting radionuclide β+, with a half-life of 110 min. When using cyclotrons to produce fluorine 18, a fraction of the quantity produced is discharged into the atmosphere in a controlled manner if the facility is not equipped with a temporary gas retention device. Most medical cyclotrons are installed in urban or peri-urban environments. Studies focusing on the radiological impact of fluorine 18 on populations living near to cyclotrons (<200 m) frequently assume a normal atmospheric distribution of concentrations

of fluorine 18 for simplification purposes. On this basis, Gaussian models are used, despite their limits, as deployment requires little input data and computing resources.

In order to estimate the performance of a Gaussian model when attempting to predict atmospheric dispersion in an urban environment, Martin et al. [1] counted 11 field campaigns carried out for cities (near-field range of a discharge point < 15 km) using passive tracers, PFCs (perfluorocarbons) and/or $SF_6$ (hexafluoride), and only four in the vicinity (<2 km) of a discharge point. The CAPITOUL [2], FLUXSAP [3] and CUTE [4] campaigns complete this inventory. Experimental data on gas dispersal are rare and inadequate for a complex environment where making general assumptions for dispersal processes for an atmospheric plume is more complex due to the presence of macro- and micro-roughness, variation in roughness, and thermal gradients caused by anthropic activities and the presence of natural and artificial materials. In addition, PFCs and $SF_6$ are fluorinated greenhouse gases, therefore it would be preferable to replace them with new passive tracers in order to protect the environment.

This study focused on atmospheric dispersion in the vicinity (<500 m) of the Beuvry hospital cyclotron (50.51404° N, 2.67255° E, France) using stable helium 4 (He) as a passive tracer for the plume of fluorine 18. Field campaigns (a total of 15 experiments) were performed in an unstable atmosphere, with wind speeds of less than 5 m s$^{-1}$. In each experiment, the helium was discharged at a constant mass flow rate (g s$^{-1}$) in the cyclotron stack air. Ambient helium concentrations (g m$^{-3}$) were measured near to the ground, at distances from the discharge point of more than 10 m, to determine atmospheric transfer coefficients *ATC* (s g$^{-1}$). The *ATC* observed were then compared with *ATC* modeled using a normal distribution plume equation determined by Briggs [5] for rural and urban environments, and by Doury [6] for the rural environment. These parametrizations were widely shared and used by the scientific community [7–13], and were established using field data, and are based on a discreet description of the atmospheric boundary layer. The parametrizations applied by Briggs depend on the Pasquill–Turner atmospheric classes [14] and the distance from the source, while Doury parametrizations depend on two stability conditions (normal diffusion and weak diffusion) and plume transfer time.

Firstly, this study demonstrates the feasibility of replacing traditional greenhouse tracers of atmospheric dispersion with helium when attempting to determine *ATC* up to 500 m downwind of a discharge point. Secondly, the performance of Briggs-rural, Briggs-urban and Doury Gaussian models was estimated in terms of predicting *ATC* and *ATCmax* observed in the prevailing weather conditions during the experiments. Thirdly, new field data acquired as part of this study were compiled with other data from the literature [2,3], acquired in urban environments and equivalent weather conditions, to propose the operational parametrization of the maximum concentrations of a Gaussian plume on the ground at distances of more than 20 m from a discharge point.

## 2. Materials and Methods

### 2.1. Experimental Campaigns and the Site

Atmospheric dispersion was studied in the vicinity (<500 m) of Beuvry hospital cyclotron (Figure 1) located on the periphery of Beuvry. The cyclotron building has a floor area of 21.5 m by 24 m, with a roof height of 8.5 m. The discharge stack (50.51404° N, 2.67255° E) rises to a height of 10.2 m. The site is topographically flat. The environment is rural to the south of the cyclotron and urban to the north. The hospital is located to the north west and a housing estate can be found to the north east, mostly comprising independent bungalows. The cyclotron is separated from the housing estate by a road contained by two 15–20 m high hedges consisting of deciduous trees, and a pile of materials several meters high. A building is located 20 m to the west south west of the cyclotron with a floor area of approximately 25 m by 30 m and a height of 10 m. The very-near-field environment of the cyclotron is:

- Bituminous, up to 100 m heading north and north west, and up to 50 m heading north east;

- Vegetation up to 100 m heading south and south west and up to 50 m heading east. A 15–20 m hedge comprising deciduous trees contains the cyclotron area in the area ranging from north east to south west, at a distance not exceeding 50 m towards the east.

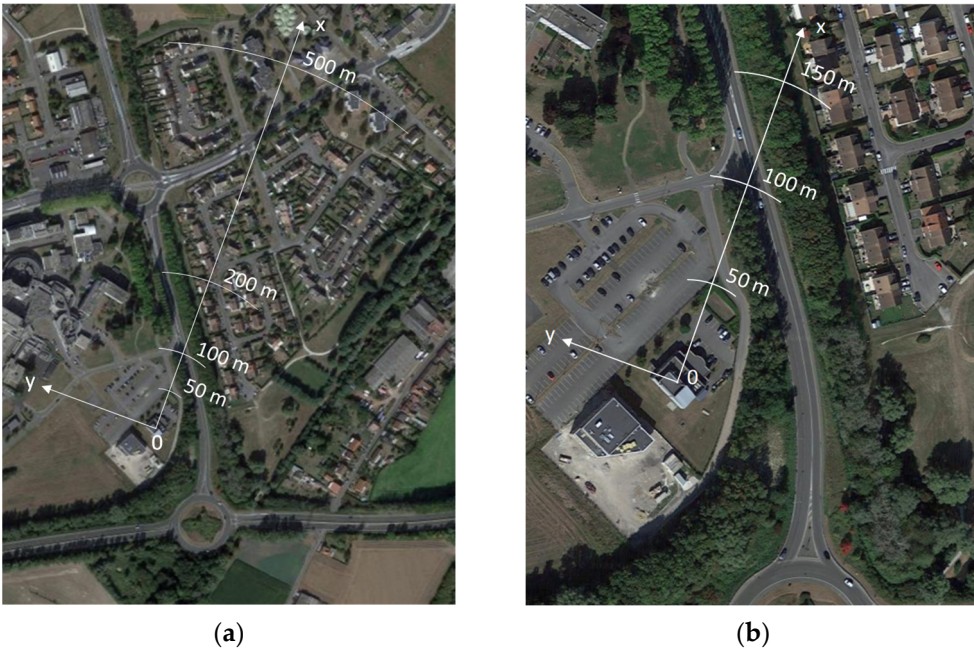

(**a**)    (**b**)

**Figure 1.** Near-field range (**a**) <500 m and (**b**) <150 m of the Beuvry hospital cyclotron (France) and distances from the discharge point with wind from 200°; the geographic coordinates of the origin shown on the images is (50.51404° N, 2.67255° E); screenshot Google Earth version 9.136.0.2 of 21 May 2021.

The geometry of the cyclotron discharge point is complex (Figure 2). The tip of the stack is curved, and the air flow is discharged at a constant flowrate of $7200 \pm 100$ m³ h⁻¹, a temperature of 20 °C and an angle of 45° downwards. Mean ejection speed is approximately 6 m s⁻¹. A roof ridge located 1 m from the stack divides the air flow into two parts. The upper part of the flow follows the roof slope, while the lower part of the flow impacts a vertical wall, inducing significant turbulence (Figure 3).

Atmospheric dispersion tracing experiments were performed around the cyclotron over 2 field campaigns, each lasting 3 days (Table 1: 15–17 October 2019 (seven experiments identified 1-*i*, *i* = 1 to 7) and 10–12 December 2019 (eight experiments identified 2-*j*, *j* = 1 to 8)). The only significant change for the site between the two campaigns relates to vegetation. The leaves of the 12–20 m hedge trees were present in October, forming a tree wall near to the cyclotron, and absent in December.

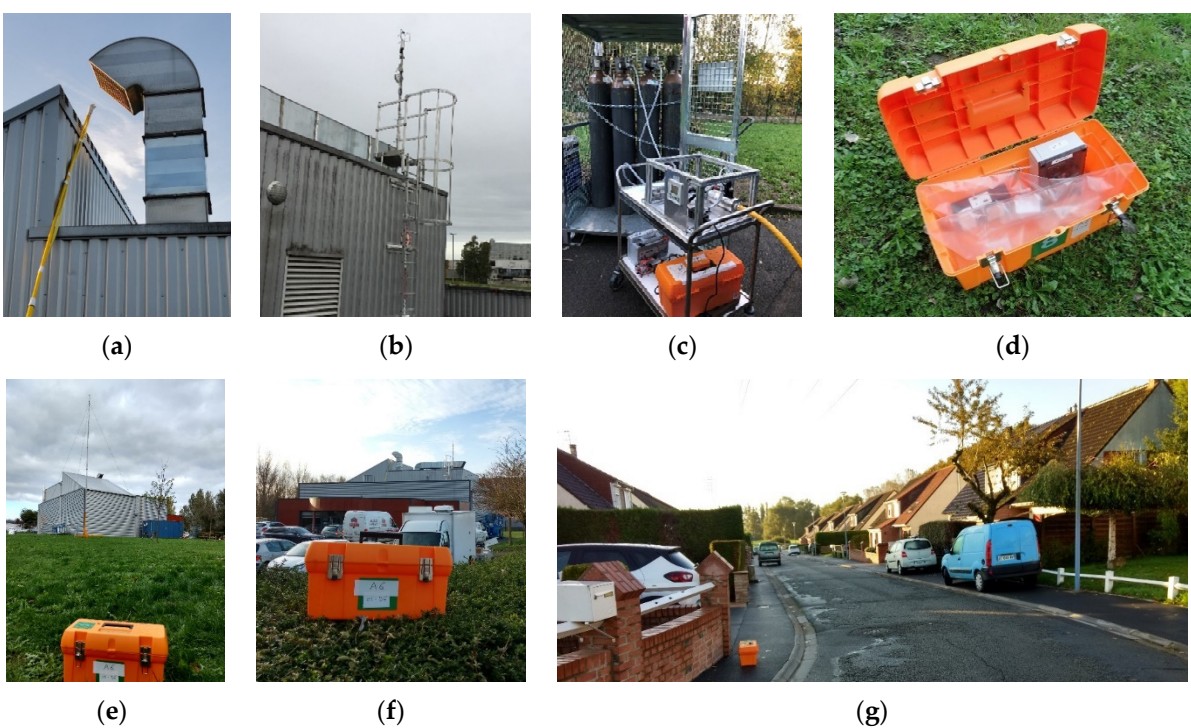

**Figure 2.** (**a**) Cyclotron discharge stack; (**b**) Young ultrasonic anemometer on the cyclotron roof; (**c**) helium discharge system; (**d**) Atmospheric sampler AS equipped with a sampling pump connected to a Tedlar bag and a flowmeter; (**e**–**g**) examples of AS locations at 40 m (**e**,**f**) and 150 m (**g**) from the discharge point.

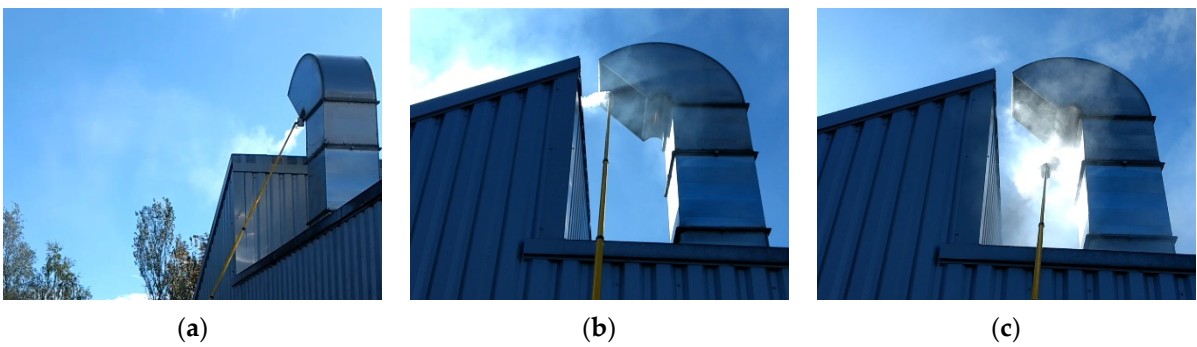

**Figure 3.** (**a**,**b**) Example of the dispersion of smoke discharged in the upper part and (**c**) in the lower part of the air flow at the stack outlet, under identical meteorological conditions.

**Table 1.** Characteristics of atmospheric helium discharges and samples for a significant helium concentration; $x$ is the distance from the discharge point in the wind direction and y is the distance perpendicular to the wind direction.

| Experiment | Date * and Time | Helium Discharge | | | Significant Sampling | | | | |
|---|---|---|---|---|---|---|---|---|---|
| | | Duration | Flowrate | Duration | Number | | $x$ | | $|y|/x$ |
| (Reference) | (UTC) | (min) | (g s$^{-1}$) | (min) | (and %) | median (m) | min (m) | max (m) | median |
| 1-1 | 15 October 2019 13:45 | 10.0 | 2.68 | 15 | 11 (79%) | 54 | 17 | 190 | 0.25 |
| 1-2 | 16 October 2019 07:10 | 10.0 | 2.38 | 15 | 11 (79%) | 29 | 12 | 67 | 1.07 |
| 1-3 | 16 October 2019 09:30 | 10.0 | 2.38 | 15 | 13 (93%) | 60 | 13 | 113 | 0.38 |
| 1-4 | 16 October 2019 12:00 | 10.0 | 5.29 | 20 | 8 (57%) | 84 | 17 | 401 | 0.22 |

**Table 1.** *Cont.*

| Experiment | Date * and Time | Helium Discharge | | | Significant Sampling | | | | | |
| | | Duration | Flowrate | Duration | Number | x | | | \|y\|/x |
| (Reference) | (UTC) | (min) | (g s$^{-1}$) | (min) | (and %) | median (m) | min (m) | max (m) | median |
| 1-5 | 17 October 2019 07:05 | 10.0 | 5.49 | 20 | 8 (57%) | 83 | 21 | 367 | 0.28 |
| 1-6 | 17 October 2019 09:30 | 10.0 | 5.36 | 15 | 13 (93%) | 151 | 35 | 279 | 0.35 |
| 1-7 | 17 October 2019 12:45 | 10.0 | 2.53 | 15 | 4 (29%) | 32 | 20 | 56 | 0.61 |
| 2-1 | 10 December 2019 09:30 | 10.0 | 2.38 | 15 | 8 (67%) | 51 | 21 | 100 | 0.49 |
| 2-2 | 10 December 2019 12:40 | 10.0 | 2.38 | 15 | 12 (100%) | 90 | 34 | 136 | 0.25 |
| 2-3 | 10 December 2019 14:20 | 10.0 | 2.38 | 15 | 10 (91%) | 86 | 34 | 133 | 0.21 |
| 2-4 | 11 December 2019 08:45 | 10.0 | 2.38 | 15 | 6 (46%) | 33 | 20 | 42 | 0.13 |
| 2-5 | 11 December 2019 10:15 | 10.0 | 2.38 | 15 | 10 (83%) | 36 | 26 | 66 | 0.17 |
| 2-6 | 11 December 2019 13:00 | 8.3 | 5.06 | 15 | 12 (92%) | 69 | 18 | 280 | 0.19 |
| 2-7 | 11 December 2019 15:00 | 9.0 | 5.06 | 15 | 10 (91%) | 267 | 38 | 502 | 0.20 |
| 2-8 | 12 December 2019 08:15 | 10.0 | 2.38 | 15 | 12 (100%) | 46 | 11 | 64 | 0.66 |

* dd/mm/yyyy.

### 2.2. Weather Conditions

Wind speed and direction were measured 7 m from the discharge point (50.51409° N, 2.67249° E), at a height of 11.8 m, by a Young ultrasonic anemometer attached to the cyclotron roof (Figure 2). Standard deviations for mean values were calculated based on instantaneous data acquired at a frequency of 10 Hz. Air temperature, atmospheric pressure and solar radiation were measured at a height of 1.5 m (50.51392° N, 2.67269° E) using a Watchdog station installed near to the cyclotron. The Pasquill–Turner atmospheric class was indirectly determined from the wind speed and solar radiation [15].

### 2.3. Tracing Experiments

A tracing experiment involves discharging helium into the atmosphere at a constant flowrate (g s$^{-1}$) for a given period and measuring the mean concentrations of helium in the plume (g m$^{-3}$), to, ultimately, calculate the atmospheric transfer coefficients *ATC* (s m$^{-3}$).

#### 2.3.1. Discharging the Passive Tracer, Helium

The helium discharge point was positioned in the air flow leaving the cyclotron stack at a height of 10.2 m. The helium discharge system comprises a B50 cylinders of pressurized helium connected to a mass flow controller (Figure 2). The discharge flowrate was constant over the discharge period and designed as a function of the distance between the discharge point and the remotest air samples. The discharge flowrate ranged between 2.4 and 2.7 g s$^{-1}$ for distances of up to 200 m from the discharge point, and between 5.1 and 5.5 g s$^{-1}$ for distances of up to 500 m (Table 1). The discharge period was equal to 10 min, except for discharges 2–6 (8.3 min) and 2–7 (9 min).

Just as PFCs and SF$_6$ were widely used in the past when studying the dispersion of gases in the atmosphere, helium is a passive tracer. The density of helium has no effect on the density of the surrounding host air, given the discharge flowrates and air renewal rate at the discharge point. Atmospheric helium concentrations measured a few meters downstream from the discharge point were around the hundred ppm.

#### 2.3.2. Air Sampling

Air samples were taken using Atmospheric Samplers AS equipped with sampling pumps connected to Tedlar bags (Figure 2). Samples were integrated to set limits on the helium plume entering the atmosphere near to the AS. Sampling begun when the helium started to be discharged into the atmosphere at the discharge point and was interrupted 15 to 20 min later based on the distance between the AS and the discharge point and wind speed. Up to 14 AS were deployed simultaneously for each experiment, including

10 equipped with a sampling flow controller (instantaneous measurement of the sampling flowrate at a frequency of 1 Hz during the sampling period). The mean sampling flowrate of an AS during an experiment was found to be 0.17 L min$^{-1}$. The median value of standard deviation for instantaneous sampling flowrates represented 3.4% of the mean sampling flowrate. Variation in the mean sampling flowrate between AS has no effect on the spatial distribution of helium concentrations. On the other hand, variation in sampling flowrate for an AS over time, during plume travel, can cause the mean concentration of helium in the atmosphere to be over- or under-estimated.

Over the 15 experiments, 194 air samples were taken downwind of the discharge point and, for each experiment, one upwind air sample was taken for control purposes to identify atmospheric background helium (Figure 2). Sample locations were selected based on wind direction, at distances $x$ from the discharge point (distances projected in the wind direction) between 10 and 500 m. A sampling height of 0.15 m was used in 92% of cases, and between 1.0 and 3.4 m in the remaining 8% of cases. The default sampling height of atmospheric samplers is 0.15 m. As this study did not aim to calculate, assess, or validate an impact on the population, this default sampling height was retained. A few samples were opportunistically taken at higher heights by positioning the samplers on urban objects or by suspending them from a meteorological mast when allowed by the wind direction.

### 2.3.3. Helium Concentrations

Helium concentrations in the air bags sampled by the Atmospheric Samplers (AS) were measured using a portable mass spectrometer ASM310 (Pfeiffer Vacuum Inc., Aßlar, Germany) equipped with a sniffer probe. The relationship between the voltage supplied by the ASM310 and the helium concentration in ppm obeys a power law. The calibration curve was established based on three reference helium concentrations: 100 ppm, 10 ppm and 5.24 ppm. The first two concentrations corresponded to the standard gases. The third concentration corresponds to the atmospheric background concentration of helium 4 [16], which is a constant. The atmospheric background concentration of helium 3, which is less than that of helium 4 by a factor of 106 [17], was ignored. The voltages supplied by the ASM310 in response to the standard concentrations were measured daily and the voltages supplied in response to atmospheric background levels were systematically measured before each concentration measurement in a bag of air. Helium concentrations were then stated in g m$^{-3}$ based on concentrations per volume (ppm), and atmospheric pressure and air temperature at the time the samples were taken.

The concentrations measured were considered significant, i.e., affected by the helium discharged during a tracing experiment, when more than 1.05 times the atmospheric background concentration of helium. The coefficient of 1.05 eliminated the influence of electronic noise from the ASM310. This coefficient was obtained by analyzing all of the measurements taken for air samples downstream from the discharge point.

The helium concentration in the plume induced by the discharge was then calculated by multiplying the difference between the significant concentration measured and the background concentration by the ratio (sampling time)/(discharge time). This correction is required as the duration of the air sampling process exceeds the duration of plume travel near to the AS. The hypothesis that the plume does not diffuse in the wind direction was verified based on 12 air samples integrated near to the ASM310 sniffer probe. The helium concentrations measured by the ASM310 at a frequency of 1 Hz were averaged over the plume life for each sample and were then compared with the concentration obtained from an integrated air sample. The relationship between the concentrations obtained using these two approaches was linear: gradient of 1.00 and a correlation coefficient $R^2$ of 0.87, for concentrations between $2.5 \times 10^{-4}$ and $4.6 \times 10^{-3}$ g m$^{-3}$. In the rest of this document and in the absence of complementary information, the default meaning of the term "helium concentration" refers to the helium concentration of the plume from the discharge point.

Being a practically inert rare gas, helium absorption by plant cover is negligible. Helium diffusion in ground porosities was also negligible as its transport time from

the discharge to the sampling points was a few minutes at most.2.3.4. Atmospheric transfer coefficients.

The wind direction is defined by the vector $\overrightarrow{Ox}$ and the height of the plume at its source $(0, 0, z_r)$ is, if no thermal convention exists, the height of the discharge point. The complexity of the cyclotron discharge point was ignored and simplified to a point discharge. Mean wind speed corresponds to that measured by the ultrasonic anemometer near to the discharge point.

The atmospheric transfer coefficient $ATC(x,y,z)$ corresponds to the ratio between the atmospheric helium concentration $C(x,y,z)$ and the helium discharge flowrate $Q$ at the discharge point (Equation (1)). At a distance $x$ from the discharge point, the $ATCmax$ modeled is obtained for $y = 0$.

$$ATC(x,y,z) = \frac{C(x,y,z)}{Q}, \tag{1}$$

where $ATC(x,y,z)$ (s m$^{-3}$) is the atmospheric transfer coefficient, $Q$ (g s$^{-1}$) the helium discharge flowrate and $C(x,y,z)$ (g m$^{-3}$) the atmospheric helium concentration.

### 2.4. Gaussian Plume Models

With a stationary plume in a spatially-uniform wind field, with no obstacles, with homogeneous and isotropic turbulence, the normal distribution of atmospheric concentrations in an axis perpendicular to the wind direction is described by Equation (2). The plume reflection on the ground is included in Equation (2) by adding of a virtual source term symmetrical to the real source with respect to the ground.

$$C(x,y,z) = \frac{Q}{2\pi\sigma_y\sigma_z\overline{u}}e^{-\frac{y^2}{2\sigma_y^2}}\left(e^{-\frac{(z-z_r)^2}{2\sigma_z^2}} + e^{-\frac{(z+z_r)^2}{2\sigma_z^2}}\right), \tag{2}$$

where $Q$ (g s$^{-1}$) is the helium discharge flowrate, $\overline{u}$ (m s$^{-1}$) is the mean wind speed, $\sigma_y$ and $\sigma_z$ (m) standard deviations for plume dispersion in the horizontal (transversal direction to the mean wind direction) et vertical directions respectively, and $z_r$ (m) the height of the discharge point.

The parametrizations proposed by Briggs for $\sigma_y$ and $\sigma_z$ were used for rural (Equation (3)) and urban (Equation (4)) environments for the Pasquill–Turner atmospheric stability conditions found during the experiments (classes B and C), and those proposed by Doury were used for the rural environment (Equation (5)). In the rest of this project, the Briggs-rural, Briggs-urban and Doury models refer to the Gaussian equation with ad hoc $\sigma_y$ and $\sigma_z$ parameters.

$$\begin{cases} \sigma_y = \{B = 0.16,\ C = 0.11\}x(1 + 0.0001x)^{-0.5} \\ \sigma_z = \{B = 0.12,\ C = 0.08\}x(1 + \{B = 0,\ C = 0.0002\}x)^{\{B=0,\ C=-0.5\}} \end{cases} \tag{3}$$

$$\begin{cases} \sigma_y = \{B = 0.32,\ C = 0.22\}x(1 + 0.0004x)^{-0.5} \\ \sigma_z = \{B = 0.24,\ C = 0.2\}x(1 + \{B = 0.001,\ C = 0\}x)^{\{B=0.5,\ C=0\}} \end{cases} \tag{4}$$

$$\begin{cases} if\ \ \frac{x}{\overline{u}} \leq 240\ s\ \ then\ \ \sigma_y = \left(0.405\frac{x}{\overline{u}}\right)^{0.859}\ and\ \ \sigma_z = \left(0.42\frac{x}{\overline{u}}\right)^{0.814} \\ if\ \ 240\ s < \frac{x}{\overline{u}} \leq 3280\ s\ \ then\ \ \sigma_y = \left(0.135\frac{x}{\overline{u}}\right)^{1.13}\ and\ \ \sigma_z = \left(\frac{x}{\overline{u}}\right)^{0.685}, \end{cases} \tag{5}$$

where $x$ (m) is the distance from the discharge point in the wind direction, $\overline{u}$ (m s$^{-1}$) is the mean wind speed, $\sigma_y$ and $\sigma_z$ (m) standard deviations for plume dispersion in the horizontal et vertical directions respectively, and B and C the Pasquill–Turner atmospheric stability classes.

A discharge time correction was applied to standard deviations for Briggs and Doury dispersion (Equation (6)) to allow for comparisons with the values obtained in this study. In

fact, the mean concentration measured at a given point in a Gaussian plume decreases with increasing sampling time, due to the presence of large atmospheric eddies. On this basis, standard deviation for plume dispersion increases with sampling time [15]. Reference sampling times used for Briggs and Doury parametrization are 30 and 6 min respectively.

$$\sigma(T_2) = \sigma(T_1) \left( \frac{T_1}{T_2} \right)^{-\alpha},\tag{6}$$

where $\sigma$ (m) is the standard deviation for plume dispersion, $T_1$ (min) the sampling time (for a plume transit time taken as equal to the duration of the discharge in this study), $T_2$ (min) the reference sampling time used by Briggs or Doury, and $\alpha = 0.5$ for $T_1 \leq 60$ min [18].

The minimum distance to the discharge point for the establishment of a Gaussian form of plume dispersion in the atmosphere is not necessarily 100 m, but depends on the topography, the integration duration of observations and weather conditions. Below 100 m, Briggs' formulas are not systematically invalid. As recalled by Griffiths [19], Briggs has accompanied his tables with graphs presenting the half-width of the plume versus the distance to the discharge point, for distances ranging from 10 to $10^4$ m. Briggs pointed out that his expressions for rural sites were good approximations of contemporary published curves for distances in the range of 100 m to $10^4$ m. However, below 100 m, the comparison was not possible due to a lack of reference studies and in situ measurements. Indeed, most of the field campaigns dealt with distances greater than 200 m (Hanford 67 and Green Glow-30 tests) or 100 m (NRTS test) [20]. Only the Prairie Grass test offered ground measurements from 50 m of the discharge point [20]. In this work, Gaussian dispersion formulas were challenged in a context at the border of their limit of validity.

### 2.5. Evaluation Criteria for Gaussian Models

The evaluation criteria described in Chang and Hanna [21] and Hanna [22] were used. For each experiment, all of the concentrations recorded (measured) and modeled were used to calculate: fractional bias *FB* (Equation (7)) the normalized mean square error *NMSE* (Equation (8)) the fraction of predictions within a factor two of observations *FAC*2 (Equation (9)) and the correlation coefficient *Corr* (Equation (10)).

$$FB = 2\frac{\overline{C_0} - \overline{C_p}}{\overline{C_0} + \overline{C_p}},\tag{7}$$

$$NMSE = \frac{\overline{(C_0 - C_p)^2}}{\overline{C_0}\,\overline{C_p}},\tag{8}$$

$$FAC2 = fraction\ of\ data\ that\ satisfies\ \ 0.5 \leq \frac{C_p}{C_0} \leq 2.0,\tag{9}$$

$$Corr = \frac{\overline{(C_0 - \overline{C_0})(C_p - \overline{C_p})}}{\sigma_{C_0}\sigma_{C_p}},\tag{10}$$

where $C_p$ (g m$^{-3}$) is the modeled concentration, $C_0$ (g m$^{-3}$) the measured concentration, and $\sigma_p$ and $\sigma_{C_o}$ (g m$^{-3}$) standard deviations for the distribution of the modeled and measured concentrations respectively.

A positive value of *FB* means that the model underestimates measurements and a negative value that the model overestimates. As reiterated in Chang and Hanna [21], a model is considered to be acceptable if $-0.3 \leq FB \leq 0.3$, $NMSE \leq 4$, $FAC2 \geq 0.5$ and $Corr \geq 0.5$.

## 3. Results

### 3.1. Weather Conditions

Mean wind direction during discharge periods was between $166 \pm 36°$ (experiments 1-2) and $233 \pm 27°$ (experiments 2-) (Table 2). For 73% of the experiments, the mean wind direction was in the same 30° angular sector, between 180° and 210°. Mean wind speed varied by a factor of 4.8 between the different experiments, ranging from $0.9 \pm 0.4$ m s$^{-1}$ (experiments 2-4) to $4.3 \pm 1.5$ m s$^{-1}$ (experiments 2-1). Standard deviation around the mean wind direction was between 19° (experiments 2-7) and 38° (experiments 1-6). In total, 25% of mean wind speeds were less than or equal to 2.0 m s$^{-1}$, 50% to 2.4 m s$^{-1}$ and 75% to 3.2 m s$^{-1}$. Standard deviation for the wind direction was between 35% (experiments 1-2) and 52% (experiments 1-2) of the mean value.

**Table 2.** Meteorological parameters.

| Experiment | Weather Conditions | | | | | |
| --- | --- | --- | --- | --- | --- | --- |
| | Wind Speed | | Wind Direction | | Solar Radiation | Stability Class |
| (Reference) | $\overline{u}$ (m s$^{-1}$) | $\sigma_U$ (m s$^{-1}$) | $\overline{Th}$ (°) | $\sigma_{Th}$ (°) | (W m$^{-2}$) | (Pasquill-Turner) |
| 1-1 | 2.5 | 1.0 | 222 | 27 | 195 | C |
| 1-2 | 2.1 | 1.1 | 166 | 36 | 14 | C |
| 1-3 | 3.3 | 1.4 | 205 | 33 | 114 | C |
| 1-4 | 3.1 | 1.4 | 210 | 35 | 78 | C |
| 1-5 | 2.0 | 0.9 | 196 | 32 | 53 | C |
| 1-6 | 2.9 | 1.1 | 186 | 38 | 201 | C |
| 1-7 | 2.1 | 1.1 | 192 | 32 | 438 | C |
| 2-1 | 3.3 | 1.3 | 183 | 24 | 118 | C |
| 2-2 | 4.3 | 1.5 | 187 | 20 | 79 | C |
| 2-3 | 4.2 | 1.9 | 184 | 24 | 22 | C |
| 2-4 | 0.9 | 0.4 | 233 | 27 | 36 | B |
| 2-5 | 1.7 | 0.7 | 197 | 19 | 110 | B |
| 2-6 | 2.0 | 0.7 | 213 | 21 | 67 | C |
| 2-7 | 2.4 | 0.9 | 195 | 19 | 4 | C |
| 2-8 | 1.7 | 0.7 | 183 | 28 | 90 | B |

The experiments were carried out during months (October and December) with low incident solar radiation in Beuvry. The atmosphere was slightly unstable (Pasquill–Turner class C), to unstable (Pasquill–Turner class B). Experiments 2-4, 2-5 and 2-8 were performed in an unstable atmosphere, with a mean wind speed of less than 2 m s$^{-1}$ (Table 2).

### 3.2. Significant Helium Concentrations

Of the 194 helium concentrations measured during 15 experiments, 148 were significant. Significant concentrations were measured at distances $x$ from the discharge point between 11 m (experiments 2-8) and 502 m (experiments 2-7) (Table 1) 25% of significant concentrations were measured less than 34 m from the discharge point, 50% less than 58 m and 75% less than 113 m. Ninety-one percent of significant concentrations were measured at a height of 0.15 m and 9% at heights between 1.0 and 3.4 m. Thirty-three percent of significant concentrations were measured in an angular sector defined as the mean wind direction $\pm 10°$ (18% with $\pm 5°$ and 6% with $\pm 2°$).

Between 11 and 14 Atmospheric Samplers AS were distributed downwind of the discharge point for each experiment. On average, significant concentrations were measured in 77% of samples (ranging from a minimum value of 29% in experiments 1-7 to a maximum value of 100% in experiments 2-2 and 2-8). A low percentage of significant concentrations frequently indicates a change of wind direction between the point in time when the AS were set in position and the start of discharging. Significant concentrations were measured for

mean ratios $|y|/x$ between 0.21 (experiments 2-6) and 1.06 (experiments 1-2). The highest ratios $|y|/x$ ($\geq 0.65$) were measured during experiments in the immediate vicinity of the discharge point, for mean distances $x$ less than 50 m (experiments 1-2, 1-7 and 2-8). During each experiment, at least one significant concentration was measured on either side of the mean wind direction.

### 3.3. Atmospheric Transfer Coefficients

Atmospheric transfer coefficients were calculated based on significant helium concentrations. For samples taken 0.15 m above the ground, *ATC* reduced by approximately two orders of magnitude depending on the distance from the discharge point (Figure 4). The *ATC* maximum value of $1.5 \times 10^{-3}$ s m$^{-3}$ was measured as $x = 21$ m from the discharge point and at $|y| = 13$ m from the wind direction (experiment 2-4), and the *ATC* minimum value of $2.0 \times 10^{-5}$ s m$^{-3}$ at $x = 401$ m from the discharge point and at $|y| = 29$ m from the wind direction (experiment 1-4). The upper values of *ATC* were limited by a power law with a power of approximately -1.3. For distances from the discharge point in excess of 140 m, the *ATC* measured in October were less than those measured in December by a factor of 3, for equivalent mean ratios $|y|/x$: $0.19 \pm 0.17$ in October and $0.15 \pm 0.12$ in December.

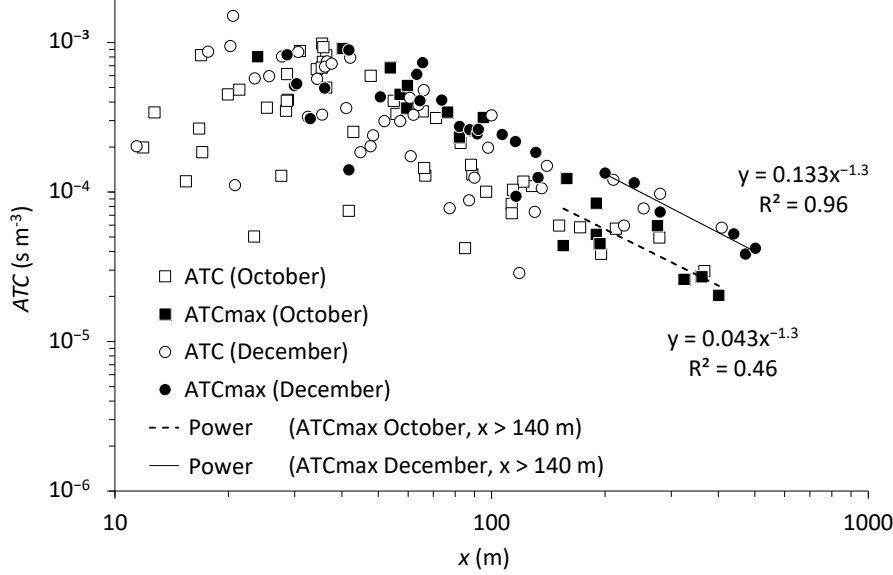

**Figure 4.** Atmospheric transfer coefficients *ATC* measured at a height of 0.15 as a function of distance $x$ from the discharge point.

### 3.4. Evaluation of Gaussian Models

Ratios $ATC_p/ATC_o$ between predicted ($p$) and observed ($o$) *ATC* were combined into six distance classes distributed logarithmically between 11 m and 502 m from the discharge point. Variation in mean ratio $ATC_p/ATC_o$ as a function of the distance from the discharge point shows that the Briggs-rural, Briggs-urban and Doury models underestimate measurements by a factor ranging from two to several orders of magnitude (Figure 5). The statistical criterion *FAC*2 is outside of the acceptability range at any distance from the discharge point (Figure 6). Statistical criteria *FB*, *NMSE* and *Corr* are within their acceptability range in the following cases: *FB* for $151 \leq x \leq 253$ m with Briggs-rural; *NMSE* pour $x \geq 40$ m with Briggs-urban and Doury, $x \geq 76$ m with Briggs-rural; *Corr* for $x \leq 21$ m with Doury, and for $76 \leq x \leq 140$ m for all of the models. According to the summary of statistical criteria, the three Gaussian models tested to model changes in *ATC* in the vicinity of the cyclotron satisfy at most two acceptability criteria out of four for $76 \leq x \leq 140$ m.

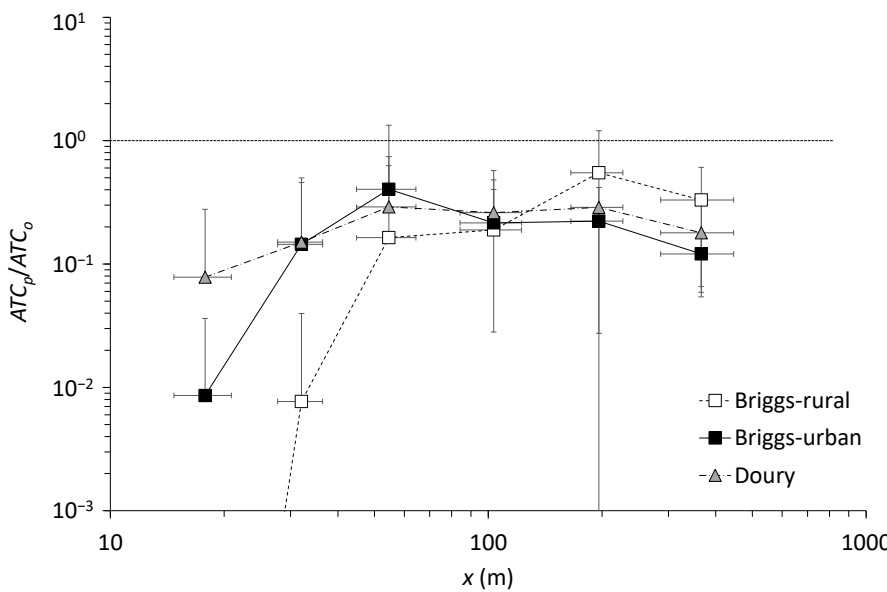

**Figure 5.** Ratio between the atmospheric transfer coefficients $ATC_p$ predicted by the Briggs-rural, Briggs-urban and Doury Gaussian models and $ATC_o$ measured as a function of distance x from the discharge point.

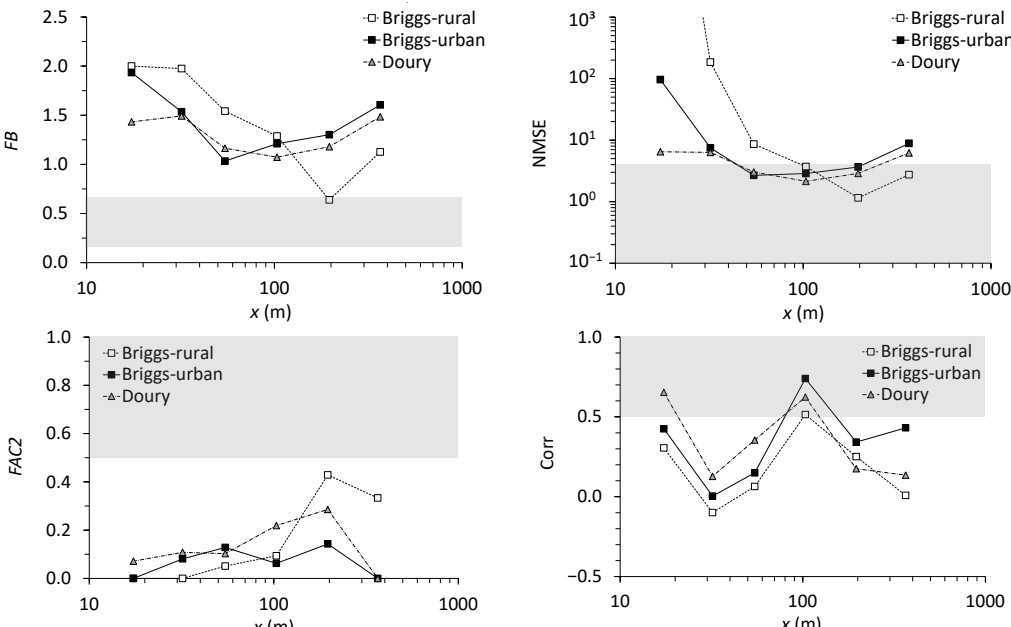

**Figure 6.** Evaluation criteria *FB*, *NMSE*, *FAC*2 and *Corr* for Briggs-rural, Briggs-urban and Doury Gaussian models when predicting the atmospheric transfer coefficients measured $CTA_o$ as a function of distance *x* from the discharge point; the acceptability zone for each criterion is shown in grey.

Ratios $ATCmax_p/ATCmax_o$ between the *ATC* maximum predicted and observed values were calculated assuming an approximation of $ATC_o$ by $ATCmax_o$ when the measured concentrations were within an angular sector defined as the main wind direction $\pm10°$. Ratios $ATCmax_p/ATCmax_o$ were combined into five distance classes distributed logarithmically between 24 m and 502 m from the discharge point. The mean ratio $ATCmax_p/ATCmax_o$ is nearer to the unit value exclusively with ratio $ATC_p/ATC_o$ (Figure 7). Statistical criteria *FB*, *NMSE*, *FAC*2 and *Corr* are within their acceptability range in the following cases (Figure 8): *FB* for $155 \leq x \leq 229$ m with Briggs-rural; *NMSE* for all models and discharge distances, except Briggs-rural for $x \leq 36$ m and Briggs-urban

for $x \geq 276$ m; $FAC2$ for $x \leq 36$ m with Briggs-urban and Doury, for $x \geq 76$ m with Briggs-rural, and for $76 \leq x \leq 239$ m with Doury; *Corr* for $76 \leq x \leq 133$ m with Briggs-urban, and for $x \geq 276$ m with Briggs-rural and Briggs-urban. According to the summary of statistical criteria, the Gaussian models tested to model changes in $ATC_{max}$ in the vicinity of the cyclotron satisfy at most three acceptability criteria out of four for $x \leq 36$ m with Briggs-urban and for $x \geq 155$ m with Briggs-rural.

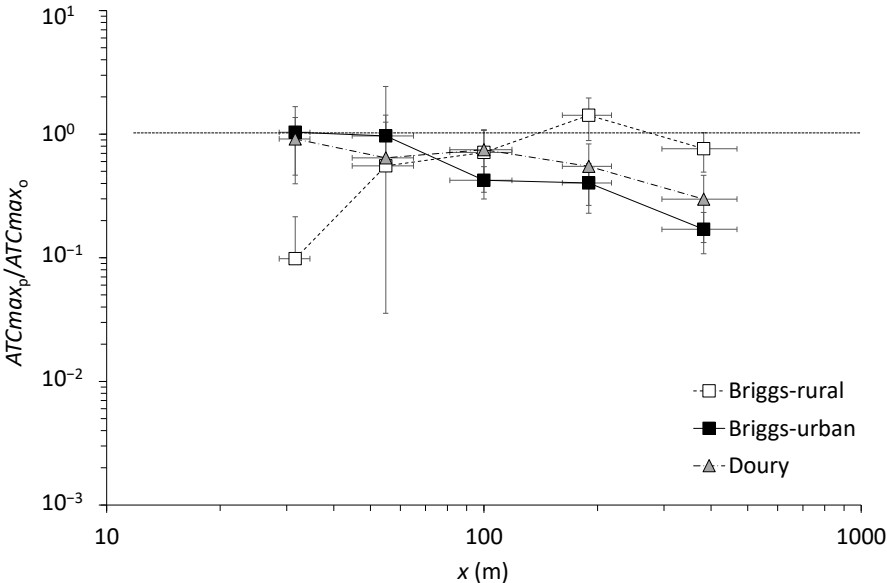

**Figure 7.** Ratio between the maximum atmospheric transfer coefficients $ATCmax_p$ predicted by the Briggs-rural, Briggs-urban and Doury Gaussian models and $ATCmax_o$ measured as a function of distance $x$ from the discharge point.

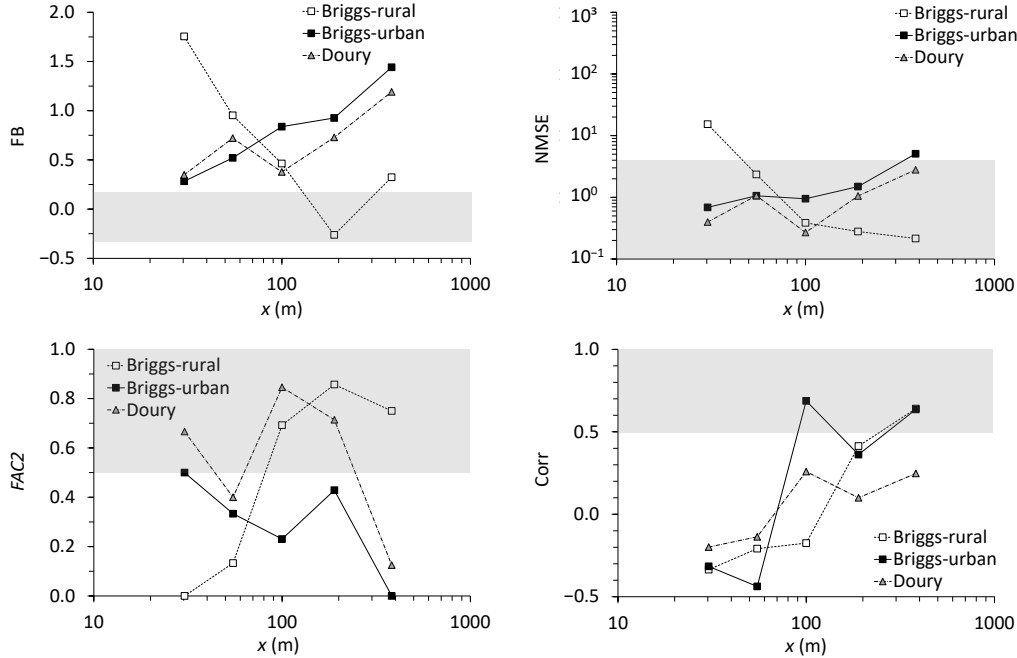

**Figure 8.** Evaluation criteria $FB$, $NMSE$, $FAC2$ and $Corr$ for Briggs-rural, Briggs-urban and Doury Gaussian models when predicting the maximum atmospheric transfer coefficients measured $CTAmax_o$ as a function of distance $x$ from the discharge point; the acceptability zone for each criterion is shown in grey.

## 4. Discussion

### 4.1. Atmospheric Transfer Coefficients Measured

At a distance $x$ from the discharge point, the dispersion of *ATC* (Figure 4) mainly depends on the distance of the Atmospheric Samplers (AS) from wind direction, and, to a lesser extent, variations in macroscopic roughness at the site. Wind directions during experiments, generally between 180° and 210°, covered angular sectors characterized by landscapes and similar changes in roughness. The minimum values of *ATC* depend on AS deployment strategy. *ATC* values are less than or equal to *ATCmax*, which is assumed to lie in the main wind direction. Thirty three percent of *ATC* considered as *ATCmax* (measured in an angular sector defined as the wind direction ±10) reduce with distance $x$ from the discharge point, for $10 \leq x \leq 500$ m, according to a power law with a power of $-1.3$ (Figure 4). *ATC* for the campaigns run in October and December 2019, measured in similar weather conditions, are identical for $x \leq 140$ m. Beyond this distance, *ATC* measured in December exceed those measured in October by a factor of two on average. This difference could be caused by seasonal vegetation effects. Based on this assumption, tree leaf-loss between October and December increased the porosity of the tree wall between the cyclotron and the housing estate, limiting plume height and leading to higher concentrations at ground level.

### 4.2. Atmospheric Transfer Coefficients Modeled

Three distance intervals $x$ with respect to the discharge point are apparent:

- The 10–50 m interval characterized by turbulent airflow in the wake of buildings and probably recirculating zones due to the fact that the cyclotron is near to a row of 15–20 m high trees;
- The 50–150 m interval with porous (two rows of trees) and non-porous (piles of materials) vertical obstacles;
- The 150–500 m interval with a housing estate of uniform roughness located at the bottom of the pile of material.

At 10–50 m from the discharge point, significant concentrations of helium were measured at the foot of the cyclotron building, indicating that the plume is almost immediately brought down to ground level. In the immediate vicinity of the discharge point, normal distribution models underestimate *ATC* by several orders of magnitude. This underestimation would be even more significant without considering that the plume rebounds from the ground, less than a few dozen meters from the discharge point. *ATCmax* modeled by Briggs-urban and Doury globally match the observations. This distance interval corresponds to the interval used by Briggs-urban to predict *ATCmax* and most matches the evaluation criteria of models: three out of four criteria are satisfied for $10 \leq x \leq 36$ m. *ATC* are underestimated on the ground, which, when combined with a satisfactory prediction of *ATCmax*, leads to the over-estimation of concentrations at altitude. The horizontal spreading of the plume was promoted by the nearby row of trees and the complexity of the discharge point, which is similar to a volume discharge flow. Considering only a simplify discharge point, it would be responsible for an underestimation of the plume width, leading to an over-estimation of concentrations in the axis of the plume and an underestimation of concentrations at the edge of the plume.

At $50-150$ m from the discharge point, the three Gaussian models tested underestimate *ATC* by a factor between 2.5 and 6.1, and *ATCmax* by a factor up to 2.5. This distance interval is a transition zone, where the initial discharge conditions have less of an influence on dispersion and where the plume spreads out more vertically. The Briggs-rural, Briggs-urban and Doury models used to predict *ATC* all respond to the evaluation criteria for the models in a similar manner: two out of four criteria at most are satisfied for $76 \leq x \leq 140$ m.

At $150-500$ m from the discharge point, the difference between measured *ATC* and modeled *ATC* remains less than one order of magnitude. *ATC* and *ATCmax* predicted by Briggs-rural models are nearer to measurements, just like the Briggs-urban and Doury models in the immediate vicinity of the discharge. This distance interval corresponds to the

interval used by Briggs-rural to predict *ATCmax* and most matches the evaluation criteria of models: three out of four criteria are satisfied for $x \geq 155$ m.

### 4.3. Parametrization of $ATC_{max}$ as a Function of the Distance x from the Discharge Point

Measured *ATCmax* in the main wind direction $\pm 10°$ were compared with *ATCmax* for Gaussian dispersion observed in an urban environment during CAPITOUL [2] and FLUXSAP [3] campaigns (Figure 9) in unstable atmospheric conditions (Pasquill–Turner classes A to C). During these campaigns, $SF_6$ was used as a tracer gas, discharge points were located on the ground, and air concentration was measured near to the ground, distributed on radials at a distance from 290 m to 5510 m (CAPITOUL campaigns), and from 59 m to 653 m (FLUXSAP campaigns) from the discharge point. Although the discharge point is high up in this study, and some assumptions based on Gaussian dispersion are not maintained, measured *ATCmax* obey the same power law as a function of distance $x$ from the discharge point as the CAPITOUL and FLUXSAP campaigns (Figure 9 and Equation (11)), crediting a Gaussian approach based on near-field observations of a discharge point ($x \geq 20$ m) in an urban environment. In Equation (11), the gradient and $x$-axis coordinate of the origin are given for a 95% confidence interval.

$$log_{10}(ATCmax) = -[0.65 \pm 0.26] - [1.55 \pm 0.11]log_{10}(x), \tag{11}$$

where *ATCmax* is the maximum atmospheric transfer coefficient (s m$^{-3}$) and $x$ (m) is the distance from the discharge point in the wind direction.

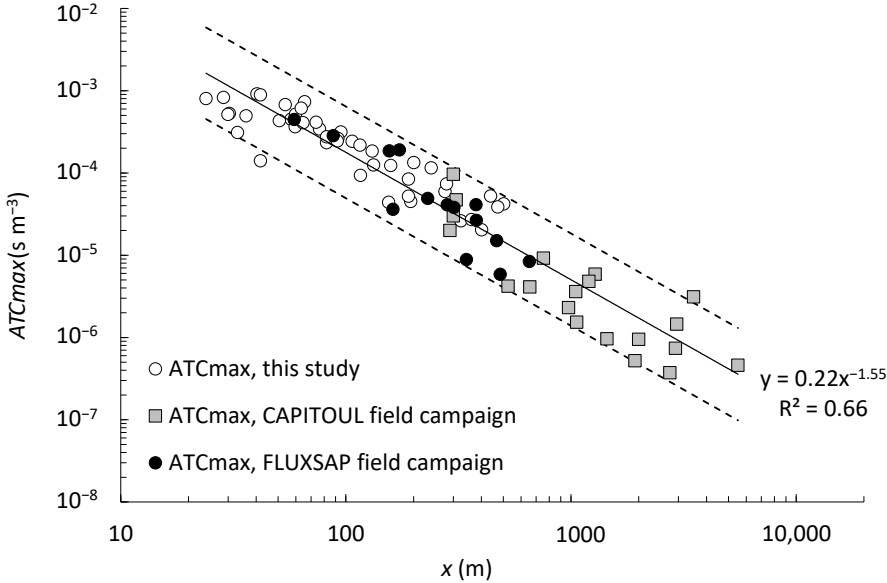

**Figure 9.** Maximum atmospheric transfer coefficients $ATC_{max}$ measured near to the ground as a function of distance $x$ from the discharge point for Pasquill–Turner atmospheric stability classes A to C; comparison with data from CAPITOUL [2] and FLUXSAP [3] campaigns; the dashed lines indicate the 95% confidence interval defined based on all data.

## 5. Conclusions

Studies focusing on the radiological impact on local populations (<200 m) near to a point where radionuclides are discharged into the atmosphere in an urban environment frequently assume a normal distribution of concentrations in the plume. In the low density urban landscapes of campaigns CAPITOUL [2], FLUXSAP [3] and this study, with housing of a moderate height and for the given weather conditions, the results obtained show that the Gaussian approach is possible when attempting to predict maximum concentrations near to the ground with a confidence interval of approximately one order of magnitude, for a distance to the discharge point in the wind direction in excess of 20 m. On the other hand,

the parametrizations used for the Briggs-rural, Briggs-urban and Doury models tested are not satisfactory in terms of predicting measurements, and values were quasi-systematically underestimated. If the concentrations and/or standard deviations for dispersion are underestimated, the dosimetric impact by inhalation will also be underestimated. By expanding this approach to fine particles considered as gases, surface deposits, which are directly proportional to air concentrations [23,24], will also be underestimated. In some high density urban environments, impact calculations could require an improved understanding of the near-field dispersion process for this type of facility. Mesh models (CFD—Computational Fluid Dynamics, type), validated by field data, could meet this requirement by characterizing atmospheric airflow around buildings at a site. In addition, this study proved that helium can potentially be used as a passive tracer of near-field dispersal around a facility.

**Author Contributions:** Conceptualization, P.L. and D.M.; methodology, P.L., O.C. (Olivier Connan), T.L.T. and D.M.; validation, P.L. and T.L.T.; formal analysis, P.L. and T.L.T.; investigation, P.L., O.C. (Olivier Connan), T.L.T., S.V., J.C., O.C. (Olivier Cazimajou), L.S. (Luc Solier), P.C.-V., L.C., I.K. and M.L.G.; data curation, P.L. and T.L.T.; writing—original draft preparation, P.L.; writing—review and editing, P.L.; visualization, P.L.; supervision, P.L.; project administration, P.L.; funding acquisition, P.L., L.S. (Lionel Soulhac), A.T. and D.M. All authors have read and agreed to the published version of the manuscript.

**Funding:** This research received no external funding.

**Institutional Review Board Statement:** Not applicable.

**Informed Consent Statement:** Not applicable.

**Data Availability Statement:** Data for figures and tables are available on request from the corresponding author.

**Acknowledgments:** The authors would like to thank IRSN, Fluidyn France SARL and Ecole Centrale de Lyon for co-funding the DIFLU research project (IRSN partnership agreement no. 21607), and Advanced Accelerator Applications (Novartis) for authorizing them to deploy equipment and take samples at the Beuvry cyclotron site. The authors would particularly like to thank Guillaume Andreolety, Romuald-Alexis Lejard and the Beuvry cyclotron team for their time and dedication, their help with preparing field campaigns and their warm welcome at the site. Finally, the authors would like to thank ENSTTI for funding advanced training on atmospheric dispersal as part of the DIFLU research project.

**Conflicts of Interest:** The authors declare no conflict of interest.

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
