# Peer review of "Investigation of a Gaussian Plume in the Vicinity of an Urban Cyclotron Using Helium as a Tracer Gas"

_atmosphere, doi:10.3390/atmos13081223_

Round 1

Reviewer 1 Report

After review of the manuscript "Investigation of a Gaussian plume in the vicinity of a urban cyclotron using helium as a tracer gas," I feel that the authors have presented a solid summary of research regarding the potential errors encountered when applying Gaussian dispersion models to short-distance applications. The findings presented by the authors are useful in regulatory and health applications.

I have only a couple of questions for the authors, hoping they can expand on a few details of the source and sampling aspects of their study:

Lines 112-113: The authors state the flow from the stack divides into two parts. I don't argue that this may be occurring, but wonder if the authors have any measurements/illustrations/references that expand on that thought. They may be useful should future studies seek to examine the role of source geometry on dispersion for which the authors' measurements could be used for comparison. This geometry may also be responsible for some of the under-prediction occurring near the facility (10-50 m range, discussed later in the paper) since I don't believe it is accounted for in the plume models.

Line 175: Could the authors provide a bit more detail on their selection of sampling heights? Most regulatory models sample at 1.5-2m (breathing height). Is there a reason the authors used 0.15m? Also, is there a reason for the variance in sampling heights, and why only a few samplers were placed higher? It may be simply a function of terrain/accessibility, but would be helpful to know.

Reviewer 2 Report

The work is, in general, interesting, but has some methodological gaps that need to be corrected,  corrections that must be made before the eventual publication of the work.

In particular:

-  The authors point out the complexity of the geometry of buildings close to the emission point, but the model they use seems not to take into account the presumably considerable effects of building downwash and stack-tip downwash. Ignoring such effects can lead to large variations in estimated concentrations, especially at the distances considered and with the wind speeds during the experiments.

- Most of the data collected were taken at distances less than 100 m. from the emissive source, in ranges where theused  Briggs sigma formulas  lose their validity.

-The used Gaussian model  seems not to consider the effect of reflection/absorption of the pollutant by the soil. This can lead to significant underestimates of predicted concentrations.

- Although the experiments were conducted from October to December, the Paquill-Turner classes considered are only of medium-high and moderate instability (B-C). Atmospheric turbulence conditions in which Gaussian models are known to be less accurate in predicting ground-level concentrations. It would be desirable to extend the experimental data under stable conditions as well (Classes D-F).

Round 2

Reviewer 2 Report

The corrections made to the text, additions and clarifications from the first version of the work, and new references  have considerably improved the quality of the work.